# Harmfully Manipulated Images Matter in Multimodal Misinformation Detection

## ABSTRACT

Nowadays, misinformation is widely spreading over various social media platforms and causes extremely negative impacts on society. To combat this issue, automatically identifying misinformation, especially those containing multimodal content, has attracted growing attention from the academic and industrial communities, and induced an active research topic named **M**ultimodal **M**isinformation **D**etection (**MMD**). Typically, existing MMD methods capture the semantic correlation and inconsistency between multiple modalities, but neglect some potential clues in multimodal content. Recent studies suggest that manipulated traces of the images in articles are non-trivial clues for detecting misinformation. Meanwhile, we find that the underlying intentions behind the manipulation, *e.g.,* harmful and harmless, also matter in MMD. Accordingly, in this work, we propose to detect misinformation by learning manipulation features that indicate whether the image has been manipulated, as well as intention features regarding the harmful and harmless intentions of the manipulation. Unfortunately, the manipulation and intention labels that make these features discriminative are unknown. To overcome the problem, we propose two weakly supervised signals as alternatives by introducing additional datasets on image manipulation detection and formulating two classification tasks as positive and unlabeled learning problems. Based on these ideas, we propose a novel MMD method, namely Harmfully Manipulated Images Matter in MMD (HAMI-M$^3$D). Extensive experiments across three benchmark datasets can demonstrate that HAMI-M$^3$D can consistently improve the performance of any MMD baselines.

## CCS CONCEPTS

• **Computing methodologies** → **Artificial intelligence**; • **Information systems** → **Social networks**.

## KEYWORDS

social media, misinformation detection, image manipulation, multimodal learning, positive and unlabeled learning

## 1 INTRODUCTION

During the past decade, prevalent social media platforms, *e.g.,* Twitter and Instagram, have bridged people from all corners of the world and made sharing information much more convenient. However, with the rise of these platforms, various misinformation has also

*ACM MM, 2024, Melbourne, Australia*
© 2024 Copyright held by the owner/author(s). Publication rights licensed to ACM.
ACM ISBN 978-x-xxxx-xxxx-x/YY/MM
https://doi.org/10.1145/nnnnnnn.nnnnnnn

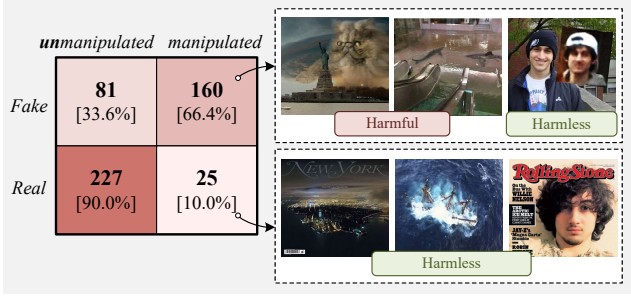

**Figure 1: The statistics on an MMD dataset *Twitter* illustrate the quantitative relationship between image manipulation and veracity labels. We use a pre-trained image manipulation detector to discriminate whether the image has been manipulated. We also provide several examples of images manipulated with harmful and harmless intentions.**

spread widely with malicious intentions, causing damages to people's mental and property [40, 42]. To eliminate such damages, the primary task becomes to automatically detect misinformation from social media, giving birth to the active topic named **M**isinformation **D**etection (**MD**).

Generally, the objective of MD is to train the veracity predictor that can automatically distinguish whether an article is real or fake. The previous arts map raw articles into a high-dimensional semantic space and learn potential correlations between these semantics and their veracity labels by designing a variety of deep models [21, 36, 55, 61]. However, most existing MD efforts solely handle text-only articles, which is unrealistic for nowadays social media platforms that contain a large amount of multimodal content. Therefore, **M**ultimodal **M**isinformation **D**etection (**MMD**) approaches have been developed recently to meet the practical need, which detects misinformation containing multiple modalities, *e.g.,* text and image. The typical MMD pipelines first capture unimodal semantic features with various prevalent feature extractors [12, 17], and then align and fuse them into a multimodal feature to predict the veracity labels [8, 15, 34, 44, 48]. Building upon this pipeline, cutting-edge MMD works design innovative multimodal interaction strategies to fuse semantic features [23, 34, 52], and learn the semantic inconsistency between different modalities [8, 32, 38].

The existing MMD methods can be beneficial from modality features, however, they also treat MMD as a standard classification problem that relies on the semantic information of samples. In contrast, misinformation is a complex phenomenon whose veracity is influenced by various aspects. As surveyed in [3, 4], a majority of fake articles may contain manipulated images created by various techniques, *e.g.,* copy-moving and splicing [9, 22]. To verify this viewpoint, we conduct a preliminary statistical analysis across a public MMD dataset *Twitter*, as shown in Fig. 1. The observation

is two-fold. On one hand, we observe that approximately 66.4% of fake articles involve manipulated images, motivating us that manipulated images can be a discriminative indicator for fake articles. On the other hand, we observe that approximately 10.0% of real articles also involve manipulated images, seeming a bit conflict to the commonsense that real articles should be completely real. We examined all those manipulated images and empirically found the ones of fake articles are more likely with harmful intentions such as deception and pranks, but the ones of real articles are with harmless intentions such as watermarking and aesthetic enhancements, as examples shown in Fig. 1. Upon these observations and the intention perspective [10], we assume that harmfully manipulated images can be a discriminative indicator for fake articles.

Motivated by these considerations, we propose to detect misinformation by extracting distinctive *manipulation features* that reveal whether the image is manipulated, as well as *intent features* that differentiate between harmful and harmless intentions behind the manipulation. Accordingly, we design a novel MMD framework, namely **HA**rmfully **M**anipulated **I**mages **M**atter in **MMD** (**HAMI-M$^3$D**). Specifically, we extract manipulation and intention features from multimodal articles and use them to formulate two binary manipulation and intention classification tasks. We then supervise these two classifiers with their corresponding binary labels. Unfortunately, their ground-truth labels are **unknown** in MMD datasets. To address this issue, we suggest the following two weakly supervised signals as substitutes for these labels. First, to supervise the manipulation classifier, we resort to a knowledge distillation paradigm [18, 24, 59] to train a manipulation teacher, which can discriminate whether an image has been manipulated, and distill its discriminative capabilities to the manipulation classifier. Specifically, we introduce additional benchmark datasets on **I**mage **M**anipulation **D**etection (**IMD**) [14] to pre-train the manipulation teacher. To further alleviate the distribution shift problem between IMD data and MMD data, we synthesize some manipulated images based on MMD datasets and formulate a **P**ositive and **U**nlabeled (**PU**) learning objective to transfer the teacher to the MD data. Second, based on the fact that *if the image of the real information has been manipulated, its intention must be harmless*, we can also formulate intention classification as a PU learning problem, and solve it by a variational PU method.

We evaluate our method HAMI-M$^3$D across 3 benchmark MMD datasets and compare it with 5 baseline MMD models. The experimental results demonstrate that HAMI-M$^3$D can improve the average performance of its baselines by approximately 1.21 across all metrics, which indicates the effectiveness of HAMI-M$^3$D. *Our source code and data will be released once the paper is accepted.*

In summary, our contributions are following three-folds:

- We suggest that image manipulation and its underlying intentions matter in MMD. To extract and integrate manipulation and intention features, we propose a new MMD model HAMI-M$^3$D.
- To solve the issue of unknown manipulation and intention labels, we propose two weakly supervised signals based on additional IMD data and PU learning.

- Extensive experiments are conducted across three MD datasets to demonstrate the improvements of HAMI-M$^3$D on the existing baseline model.

## 2 RELATED WORKS

In this section, we briefly review and introduce the related literature on MMD, manipulation detection, and PU learning.

### 2.1 Multimodal Misinformation Detection

In general, the primary goal of MMD is to automatically distinguish misinformation consisting of text-image pairs on social media. Most existing MMD methods concentrate on creating powerful multimodal models to grasp complex semantic information [34, 44, 48]. For example, BMR [52] refines and fuses multimodal feature using an improved mixture-of-experts network. In addition, several MMD arts suggest modeling the inconsistency between different modalities [8, 15], and they present a hypothesis that the inconsistency between modalities is a non-trivial clue for detecting misinformation. Following this hypothesis, CAFE [8] proposes a variational approach to calculate the inconsistency between modalities and leverage it to guide the multimodal feature fusion. Our study aims to improve MMD models by specifying manipulation and its intent features. Despite the importance of the image manipulation in misinformation detection, few works have explored the role of manipulation features. Previous studies [1, 3, 4, 28, 33] have intuitively highlighted the significance of such features, but specific models have not been developed yet.

### 2.2 Manipulation Detection

With the rapid development and popularization of multimedia technology, it has become increasingly easy to manipulate various multimedia contents, *e.g.,* images and videos, through copy-moving [9], splicing [22] or inpainting [27]. To automatically control the misuse of these techniques, IMD has become an active topic in the information forensics community [10, 35]. Briefly, The primary goal of IMD is to determine if an image has been manipulated and localize the specific manipulated area, which is a challenging segmentation task. Most of the existing methods focus on designing powerful neural models to extract effective semantic features and capture subtle manipulation traces [30, 39, 57]. Apart from network structures, some works are also dedicated to designing more efficient training strategies. For example, some pre-training datasets are synthesized by manipulating real images to extend the tampered class [49, 50, 58]. Motivated by these arts, we design a hierarchical visual encoder as the basic manipulation teacher to obtain multi-view manipulation features, pre-train it with large-scale IMD datasets, and adapt the model to the MMD task with synthesized PU learning.

### 2.3 Positive-Unlabeled Learning

PU learning is a unique learning paradigm that aims to learn a binary classifier by accessing only a part of labeled positive samples and several unlabeled samples. Generally, existing PU learning methods are summarized into two basic categories, sample-selection and cost-sensitive methods. The sample-selection line leverages several heuristic strategies to select potential negative samples from the unlabeled data, and then train the binary classifier

with the supervised or semi-supervised learning paradigm [20, 53]. For example, PULUS [29] learns a negative sample selector with reinforcement learning by the reward from the performance across the validation set. On the other hand, the cost-sensitive line designs a variety of empirical risks on negative samples and constrains them unbiased [6, 16, 25, 56]. For example, uPU [16] early proposes to unbiasedly estimate risks. Additionally, some current methods attempt to assign reliable pseudo labels to unlabeled samples [19, 29, 45] and design effective augmentation methods [26, 47].

In the MD community, some recent works also solve the PU misinformation detection topic [11, 43]. They present a weakly-supervised task, which learns misinformation detectors with partial real articles as the positive samples and regards the other articles as unlabeled samples. Unlike them, our HAMI-M$^3$D employs PU learning as an important tool to adapt the pre-trained IMD model to MMD datasets and resolve the issue of unknown intention labels.

# 3 PROPOSED HAMI-M$^3$D METHOD

In this section, we will introduce the proposed MMD model **HAMI-M$^3$D** in more detail.

**Problem definition.** Formally, an MMD dataset typically contains $N$ training samples $\mathcal{D} = \{(\mathbf{x}_i^T, \mathbf{x}_i^I, y_i)\}_{i=1}^N$, where $\mathbf{x}^T$ and $\mathbf{x}^I$ are respectively the text content and images of an article, and $y \in \{0, 1\}$ denotes the corresponding veracity label (0/1 indicates real/fake). The basic target of MMD is to learn a detector to predict the veracity of any unseen article. The basic framework of the existing MMD method typically consists of three modules: feature encoder, feature fusion network, and predictor. The feature encoder extracts the unimodal semantic features of $\mathbf{x}_i^T$ and $\mathbf{x}_i^I$. The feature fusion network integrates these features into a multimodal feature before feeding into the predictor module.

## 3.1 Overview of HAMI-M$^3$D

We are inspired by the joint observation and assumption that fake articles may be highly relevant to harmfully manipulated images. Accordingly, for each sample, we can extract its latent manipulation feature and harmful feature and then fuse them with the semantic features to achieve more discriminative fused features. To estimate the two latent features, we conduct the manipulation classification and harmful classification as auxiliary tasks. Upon these ideas, we design HAMI-M$^3$D under a multi-task learning framework jointly learning the primary task of veracity classification with the two auxiliary tasks. To be specific, HAMI-M$^3$D consists of three primary modules: **feature encoders module**, **feature fusion module**, and **predictors module**. For clarity, the overall framework of HAMI-M$^3$D is illustrated in Fig. 2. In the following, we introduce these modules in more detail.

**Feature encoders module.** This module consists of four specific feature encoders, including text encoder, image encoder, manipulation encoder, and intention encoder.

Given a pair of text content $\mathbf{x}_i^T$ and an image $\mathbf{x}_i^I$, the text encoder and image encoder extract their respective text and image features $\mathbf{e}_i^T$ and $\mathbf{e}_i^I$. Specifically, we derive text and image features $\mathbf{e}_i^T = \mathcal{F}_{\Theta^T}(\mathbf{x}_i^T)$ and $\mathbf{e}_i^I = \mathcal{F}_{\Theta^I}(\mathbf{x}_i^I)$, respectively, by leveraging a pre-trained BERT model [12] and a ResNet34 model [17]. These

features are then aligned into a shared feature space using two feed-forward neural networks. Next, we directly input the image feature $\mathbf{e}_i^I$ into a manipulation encoder to generate the manipulation feature $\mathbf{e}_i^M = \mathcal{F}_{\Theta^M}(\mathbf{e}_i^I)$. Based on this manipulation feature, we then integrate it with the text and image features $\mathbf{e}_i^T$ and $\mathbf{e}_i^I$ into an intention encoder to obtain the intention feature $\mathbf{e}_i^E = \mathcal{F}_{\Theta^E}(\mathbf{e}_i^T, \mathbf{e}_i^I, \mathbf{e}_i^M)$.

**Feature fusion module.** Given these extracted features, the feature fusion module utilizes a multi-head attention network to integrate them into one fused feature $\mathbf{z}_i = \mathcal{F}_{\Psi^F}(\mathbf{e}_i^T, \mathbf{e}_i^I, \mathbf{e}_i^M, \mathbf{e}_i^E)$.

**Predictors module.** This module contains three predictors trained on three different tasks: veracity classification, manipulation classification, and intention classification. Utilizing the fused feature $\mathbf{z}_i$, a linear veracity classifier is employed to predict the veracity label as $p_i = \mathbf{W}_V \mathbf{z}_i$. The objective for the veracity classification task across $\mathcal{D}$ can be formulated as follows:

$$\mathcal{L}_{VC} = \frac{1}{N} \sum_{i=1}^N \ell_{CE}(p_i, y_i), \tag{1}$$

where $\ell_{CE}(\cdot, \cdot)$ denotes a cross-entropy loss function. Then, given manipulation and intention features $\mathbf{e}_i^M$ and $\mathbf{e}_i^E$, we deploy manipulation and intention classifiers to generate their corresponding predictions $p_i^M = \mathbf{W}_M \mathbf{e}_i^M \in [0, 1]$ and $p_i^E = \mathbf{W}_E \mathbf{e}_i^E \in [0, 1]$, where $p_i^M = 1$ or 0 indicates whether the image has been manipulated or not, and $p_i^E = 1$ or 0 represents the harmless or harmful intention behind the manipulation, respectively.

Unfortunately, the ground-truth manipulation and intention labels are **unknown** in MMD datasets. To overcome this issue, we introduce two weakly supervised signals as substitutes to achieve the manipulation classification and intention classification tasks. Specifically, for manipulation classification, we train a manipulation teacher $f_{\Pi}(\cdot)$ using additional IMD datasets, *e.g., CASIAv2* [14], with an objective $\mathcal{L}_{PRE}$. To address the distribution shift issue between IMD datasets and MMD datasets, we then adapt the teacher by utilizing a PU loss $\mathcal{L}_{PU}$. Given the teacher's output $y_i^M = f_{\Pi}(\mathbf{x}_i^I)$, we can distill it to the prediction $p_i^M$ as follows:

$$\mathcal{L}_{KD} = \frac{1}{N} \sum_{i=1}^N D_{KL}(y_i^M, p_i^M), \tag{2}$$

where $D_{KL}(\cdot, \cdot)$ denotes the Kullback-Leibler divergence function. For intention classification, we draw inspiration from the fact that *if the image of the **real** article is manipulated, its intention must be **harmless***, and reformulate the intention classification into a PU learning problem, which can be specified by an objective $\mathcal{L}_{IR}$. Additionally, based on another fact that *if the image of one article is manipulated with a **harmful** intention, the veracity label of this article must be **fake***, we can also check the reliability of $p_i^E$, and filter out unreliable samples during training.

Based on the aforementioned tasks, our overall objectives are as follows:

$$\mathcal{L} = \mathcal{L}_{VC} + \alpha \mathcal{L}_{KD} + \beta \mathcal{L}_{IR}, \tag{3}$$

$$\mathcal{L}_{\text{teacher}} = \mathcal{L}_{PRE} + \delta \mathcal{L}_{PU}, \tag{4}$$

where $\alpha$, $\beta$, and $\delta$ are trade-off hyper-parameters to balance multiple loss functions. We will alternatively optimize our MMD model and the teacher model with the objectives in Eqs. (3) and (4). For clarity, the overall training pipeline is described in Alg. 1. In the following

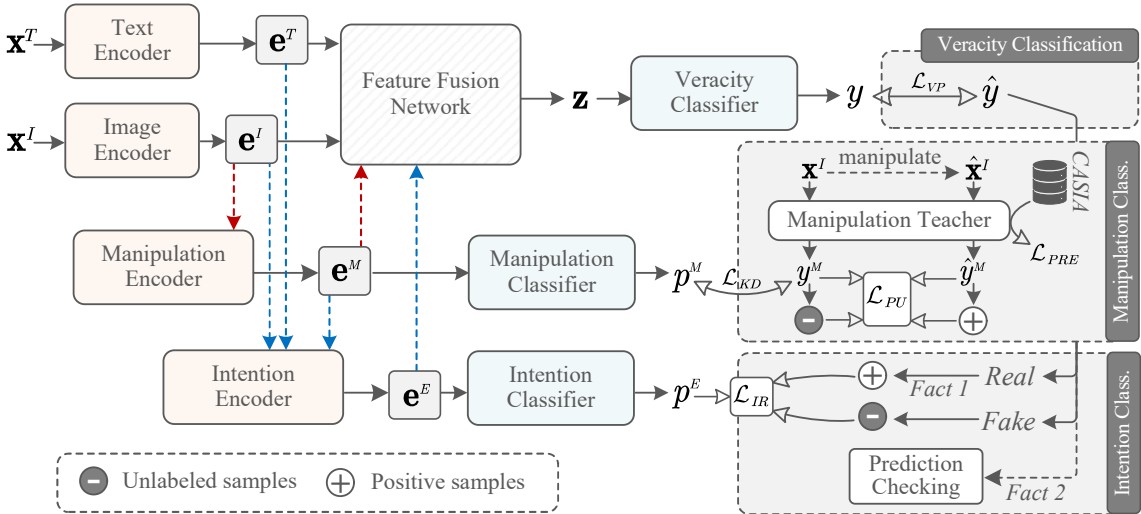

**Figure 2: The overall framework of HAMI-M³D. Given text content $x_i^T$ and an image $x_i^I$, we use four encoders including text encoder, image encoder, manipulation encoder, and intention encoder to extract their corresponding features. These features are then input into a feature fusion network to obtain a fused feature. Finally, we propose three predictors to achieve three different tasks: veracity classification, manipulation classification, and intention classification.**

sections, we introduce the manipulation and intention classification tasks in more detail.

## 3.2 Manipulation Classification

Generally, the manipulation classification task involves training a manipulation teacher $f_\Pi(\cdot)$ and then distilling its predictions to the prediction $q_i^M$ with Eq. (2). To be specific, the optimization of the teacher model requires two objectives: a pre-training objective $\mathcal{L}_{PRE}$ and an objective $\mathcal{L}_{PU}$ to adapt the model.

First, we introduce a benchmark IMD dataset, denoted as $\mathcal{D}_\mu = \{x_j^\mu, y_j^\mu\}_{j=1}^{N_\mu}$, e.g., CASIAv2 [14]. The dataset consists of $N_\mu$ pairs of images $x^\mu$ and their corresponding ground-truth manipulation labels $y^\mu \in \{0, 1\}$, where $y^\mu = 1$ or 0 indicates $x^\mu$ is manipulated or not. Given $x_j^\mu$, we feed it into a ResNet18 as the backbone of our teacher model. Since recent some IMD arts suggest that the detection of manipulation trace requires not only semantic information but also subtle clues in images [13, 39]. Therefore, we follow Chen et al. [7] to extract the features of the inner layers of ResNet18, integrate them with a self-attention network, and predict the manipulation label of $x_j^\mu$. Its objective can be formulated as follows:

$$\mathcal{L}_{PRE} = \frac{1}{N_\mu} \sum_{j=1}^{N_\mu} \ell_{CE} \left( f_\Pi(x_j^\mu), y_j^\mu \right). \quad (5)$$

Then, due to the inevitable distribution shift problem between the IMD dataset $\mathcal{D}_\mu$ and the MMD dataset $\mathcal{D}$, we propose to adapt the teacher model pre-trained across $\mathcal{D}_\mu$ to the MMD dataset through a PU learning framework. Specifically, given an image $x_i^I$ sampled from $\mathcal{D}$, we randomly manipulate it with the copy-moving approach [9], and synthesize its manipulated version $\hat{x}_i^I$. Therefore, the manipulation label of $\hat{x}_i^I$ can be naturally assigned as "*manipulated*", denoted as $\hat{y}_i^M = 1$, and form a training subset

$\mathcal{P}^M = \{\hat{x}_i^I, \hat{y}_i^M = 1\}_{i=1}^N$. Meanwhile, the manipulation label of $x_i^I$ is still unknown, so we form another unlabeled subset $\mathcal{U}^M = \{x_i^I\}_{i=1}^N$. Accordingly, drawing inspiration from PU learning, which aims to train a binary classifier with partially labeled positive samples and sufficient unlabeled samples, we reformulate the manipulation classification across $\mathcal{P}^M \cup \mathcal{U}^M$ as a PU learning problem.

Formally, several arts specify PU learning with various risk estimation approaches. In this work, we implement it with a variational PU learning framework [5].[1] Given two subsets $\mathcal{P}^M \sim \mathbb{P}_P \triangleq \mathbb{P}(x^I|y^M = 1)$ and $\mathcal{U}^M \sim \mathbb{P}_U \triangleq \mathbb{P}(x^I)$,[2] using a Bayes rule, we can estimate $\mathbb{P}_P$ as follows:

$$\mathbb{P}_P = \frac{\mathbb{P}(y^M = 1|x^I)\mathbb{P}(x^I)}{\int \mathbb{P}(y^M = 1|x^I)\mathbb{P}(x^I)dx^I} = \frac{f_{\Pi^\star}(x^I)\mathbb{P}_U}{\mathbb{E}_{\mathbb{P}_U}\left[f_{\Pi^\star}(x^I)\right]}$$

$$\approx \frac{f_\Pi(x^I)\mathbb{P}_U}{\mathbb{E}_{\mathbb{P}_U}\left[f_\Pi(x^I)\right]} \triangleq \mathbb{P}_\Pi, \quad (6)$$

where $\mathbb{P}_\Pi$ is the data distribution with the parametric model $\Pi$, and $f_{\Pi^\star}(\cdot)$ represents an optimal teacher model. Accordingly, to optimize $f_\Pi(\cdot)$ towards the optimal $f_{\Pi^\star}(\cdot)$, prior studies [5] prove that we can minimize the KL divergence between $\mathbb{P}_P$ and $\mathbb{P}_\Pi$, which

---

[1] The variational PU learning follows a "*selected completely at random*" assumption that does not require additional class priors, which is consistent with the scenario of our paper. Additionally, it has empirically demonstrated superior performance in HAMI-M³D.

[2] Since $\mathbb{P}(\hat{x}^I)$ and $\mathbb{P}(x^I)$ are independently and identically distributed, to keep our notations clear, we uniformly utilize $x^I$ and $y^M$ to indicate the image and its manipulation label.

---

**Algorithm 1** Training summary of HAMI-M$^3$D.

---

**Input:** Training MMD dataset $\mathcal{D}$; IMD dataset $\mathcal{D}_\mu$; hyper-paramaters $\alpha$, $\beta$, and $\delta$; training iterations $I$.

**Output:** An MMD model parameterized by $\Theta$; teacher model $\Pi$.

1: Initialize $\Theta^T$ and $\Theta^I$ with their pre-trained weights, and other parameters from scratch.
2: Warm-up $\Pi$ with $\mathcal{L}_{PRE}$ for 10 epochs.
3: **for** $i = 1, 2, \cdots, I$ **do**
4:   Draw mini-batches $\mathcal{B}$, $\mathcal{B}_\mu$ from $\mathcal{D}$, $\mathcal{D}_\mu$ randomly.
5:   Manipulate images in $\mathcal{B}$ and form a manipulated $\widehat{\mathcal{B}}$.
6:   Calculate $\mathcal{L}_{PRE}$ with $\mathcal{B}_\mu$ and $\mathcal{L}_{PU}$ with $\mathcal{B} \cup \widehat{\mathcal{B}}$.
7:   Optimize $\Pi$ with Eq. (4).
8:   Calculate $\mathcal{L}_{VC}$, $\mathcal{L}_{KD}$, and $\mathcal{L}_{IR}$ with $\mathcal{B}$.
9:   Optimize $\Theta$ with Eq. (3).
10: **end for**

---

is formalized as follows:

$$D_{KL}\left(\mathbb{P}_P \| \mathbb{P}_\Pi\right) = \mathbb{E}_{\mathbb{P}_P}\left[\log \frac{\mathbb{P}_P(\mathbf{x}^I)}{\mathbb{P}_\Pi(\mathbf{x}^I)}\right] \tag{7}$$

$$= \mathbb{E}_{\mathbb{P}_P}\left[\log f_{\Pi^\star}(\mathbf{x}^I)\right] + \mathbb{E}_{\mathbb{P}_P}\left[\log \mathbb{P}_P(\mathbf{x}^I)\right] - \log \mathbb{E}_{\mathbb{P}_U}\left[f_{\Pi^\star}(\mathbf{x}^I)\right]$$

$$- \left(\mathbb{E}_{\mathbb{P}_P}\left[\log f_\Pi(\mathbf{x}^I)\right] + \mathbb{E}_{\mathbb{P}_P}\left[\log \mathbb{P}_P(\mathbf{x}^I)\right] - \log \mathbb{E}_{\mathbb{P}_U}\left[f_\Pi(\mathbf{x}^I)\right]\right).$$

Accordingly, the PU optimization objective can be specified as follows:

$$\mathcal{L}_{PU} \triangleq \log \mathbb{E}_{\mathcal{U}^M \sim \mathbb{P}_U}\left[f_\Pi(\mathbf{x}^I)\right] - \mathbb{E}_{\mathcal{P}^M \sim \mathbb{P}_P}\left[\log f_\Pi(\mathbf{x}^I)\right]. \tag{8}$$

By optimizing these two objectives $\mathcal{L}_{PRE}$ and $\mathcal{L}_{PU}$ with Eq. (4), we can obtain a strong manipulation teacher, and distill the prediction $y_i^M$ from the teacher to $p_i^M$ with Eq. (2) [18, 24, 59]. Note that during the optimization process, we first use $\mathcal{L}_{PRE}$ to warm up the teacher model for 10 epochs to prevent cold start problems in the optimization of $\mathcal{L}_{PU}$.

### 3.3 Intention Classification

Given the intention feature $\mathbf{e}_i^E$, the intention classification task aims to make it discriminative about the intention behind the image manipulation. To solve the problem of unknown intention labels, we provide two facts as weakly supervised signals to supervise the intention prediction $p_i^E$. Specifically, the first fact is presented as:

**Fact 1.** *If the image of the **real** article is manipulated, its intention must be **harmless**; But if the image of the **fake** article is manipulated, its intention may be **harmful** or **harmless**.* Written as:

$$y_i^E = \begin{cases} 1, & y_i = 0 \wedge y_i^M = 1, \\ 0 \text{ or } 1, & y_i = 1 \wedge y_i^M = 1, \end{cases}$$

where $y_i^E$ denotes the intention label of the $i$-th sample.

Based on this fact, we can form a subset $\mathcal{D}^E \in \mathcal{D}$ where all samples satisfy $y^M = 1$, and then split $\mathcal{D}^E$ into a positive subset $\mathcal{P}^E$ where $y = 0$ and an unlabeled subset $\mathcal{U}^E$ where $y = 1$. Accordingly, the intention classification across $\mathcal{P}^E \cup \mathcal{U}^E$ can also be reformulated as a PU learning problem. Similar to the formula in Eq. (8), its objective is represented as:

$$\mathcal{L}_{IR} \triangleq \log \mathbb{E}_{\mathcal{U}^E \sim \mathbb{P}_U}[p^E] - \mathbb{E}_{\mathcal{P}^E \sim \mathbb{P}_P}[\log p^E]. \tag{9}$$

**Table 1: Statistics of three prevalent MMD datasets.**

| Dataset | # Real | # Fake | # Images |
|---------|--------|--------|----------|
| *GossipCop* [37] | 10,259 | 2,581 | 12,840 |
| *Weibo* [23] | 4,779 | 4,749 | 9,528 |
| *Twitter* [1] | 6,026 | 7,898 | 514 |

In addition, another fact is presented as:

**Fact 2.** *If the image of one article is manipulated by a **harmful** intention, the veracity label of this article must be **fake**; But if the image of one article is manipulated by a **harmless** intention, the veracity label of this article may be **real** or **fake**.* Written as:

$$y_i = \begin{cases} 1, & y_i^E = 1 \wedge y_i^M = 1, \\ 0 \text{ or } 1, & y_i^E = 0 \wedge y_i^M = 1. \end{cases}$$

This fact can be regarded as a metric to check the reliability of predictions $p_i^M$ and $p_i^E$. For a sample that satisfies $p_i^M = 1$ and $p_i^E = 1$, if its ground-truth veracity label $y_i \neq 1$, at least one of its predictions $p_i^M$ and $p_i^E$ is incorrect, and we remove these incorrect samples when optimizing the model with Eq. (3).

## 4 EXPERIMENTS

In this section, we conduct extensive experiments and compare HAMI-M$^3$D to existing MMD baselines to evaluate its performance.

### 4.1 Experimental Settings

**Datasets.** To evaluate the empirical performance of HAMI-M$^3$D, we conduct our experiments across three prevalent MMD datasets *GossipCop* [37], *Weibo* [23], and *Twitter* [1, 2], their statistics are shown in Table 1. Specifically, *GossipCop* and *Weibo* consist of 12,840 and 9,528 text-image pairs, respectively, and their text and images typically have a one-to-one correspondence. Differently, the *Twitter* dataset contains 13,924 texts but only 514 images, so it exhibits a complex one-text-to-many-images or one-image-to-many-texts correspondence, which is a more challenging scenario. We follow previous works [46, 54] to process and split these datasets into training, validation, and test subsets by a ratio of 7:1:2.

**Baselines.** We compare five MMD baselines and their improved versions using HAMI-M$^3$D in our experiments. These baselines are briefly introduced as follows:

- **Basic model** extracts text and image features with a pre-trained BERT model [12] and a ResNet34 [17], respectively. Then, we use typical FFNN layers to map and align two features into a shared space, and then fuse multimodal features and predict veracity labels with MLP layers.
- **SAFE** [60] designs a similarity-aware multimodal fusion module for MMD.
- **MCAN** [51] proposes a multimodal co-attention network to fuse multimodal features effectively and considers inter-modality correlations.
- **CAFE** [8] guides the multimodal feature fusion by assigning adaptive weights with a variational method, and aligns unimodal features with a contrastive learning regularization.

**Table 2: Experimental results of Hami-m³d across three prevalent MD datasets *GossipCop, Weibo,* and *Twitter*. The results marked by * are statistically significant compared to its baseline models, satisfying p-value < 0.05.**

| Method | Accuracy | Macro F1 | Real | | | Fake | | | Avg. Δ |
|---|---|---|---|---|---|---|---|---|---|
| | | | Precision | Recall | F1 | Precision | Recall | F1 | |
| **Dataset: *GossipCop*** | | | | | | | | | |
| Basic model | 87.77±0.56 | 79.51±0.44 | 91.55±0.41 | 93.36±1.20 | 92.37±0.41 | 69.96±1.10 | 63.30±1.46 | 66.92±0.58 | - |
| Basic model + Hami-m³d | 88.45±0.20* | 80.32±0.43* | 91.93±0.14 | 94.08±0.33* | 92.83±0.12 | 71.99±0.80* | 64.59±0.74* | 67.63±0.78* | +0.90 |
| SAFE [60] | 87.78±0.31 | 79.22±0.49 | 91.22±0.30 | 93.34±0.47 | 92.37±0.20 | 70.66±1.32 | 63.12±1.50 | 66.66±0.84 | - |
| SAFE + Hami-m³d | 88.53±0.24* | 79.87±0.30* | 91.90±0.31* | 94.32±0.54* | 92.95±0.20* | 72.19±1.30* | 64.44±0.73* | 67.88±0.51* | +0.96 |
| MCAN [51] | 87.66±0.59 | 78.89±0.34 | 90.89±0.78 | 94.07±1.27 | 92.19±0.46 | 71.01±1.09 | 60.37±1.21 | 65.29±0.87 | - |
| MCAN + Hami-m³d | 88.27±0.57* | 79.87±0.36* | 91.72±0.35* | 95.13±1.21* | 93.05±0.41* | 72.69±0.96* | 62.64±1.21* | 66.65±0.32* | +1.21 |
| CAFE [8] | 87.40±0.71 | 79.51±0.61 | 91.07±0.25 | 93.84±1.28 | 92.16±0.50 | 71.60±1.39 | 61.16±1.10 | 66.24±0.72 | - |
| CAFE + Hami-m³d | 88.18±0.44* | 80.43±0.48* | 91.50±0.45 | 94.46±1.00* | 92.80±0.31* | 72.84±0.83* | 62.51±0.90* | 67.58±0.83* | +0.91 |
| BMR [52] | 87.26±0.46 | 79.03±0.64 | 90.89±0.24 | 93.99±0.59 | 92.14±0.29 | 71.15±1.23 | 60.37±1.21 | 65.51±1.01 | - |
| BMR + Hami-m³d | 87.95±0.27* | 79.99±0.57* | 91.40±0.51* | 94.73±0.75* | 93.14±0.19* | 72.26±0.73* | 62.94±0.89* | 66.80±1.09* | +1.11 |
| **Dataset: *Weibo*** | | | | | | | | | |
| Basic model | 90.87±0.34 | 90.75±0.34 | 91.08±0.23 | 90.17±0.85 | 90.62±0.40 | 90.87±0.70 | 91.41±0.28 | 91.29±0.29 | - |
| Basic model + Hami-m³d | 91.62±0.66* | 91.61±0.66* | 91.83±0.87* | 93.23±0.56* | 91.39±0.76* | 92.52±0.89* | 91.87±0.64 | 91.84±0.62* | +1.11 |
| SAFE [60] | 91.06±0.88 | 91.04±0.89 | 91.09±1.25 | 90.51±0.90 | 90.73±1.04 | 91.27±0.78 | 91.57±1.14 | 91.36±0.85 | - |
| SAFE + Hami-m³d | 92.22±0.91* | 92.22±0.93* | 91.15±1.08 | 94.22±0.84* | 92.14±0.92* | 94.34±1.00* | 91.34±1.09 | 92.30±0.66* | +1.42 |
| MCAN [51] | 90.99±0.83 | 90.99±0.83 | 89.66±0.82 | 92.24±1.10 | 90.81±0.90 | 92.69±0.80 | 89.92±0.99 | 91.20±0.79 | - |
| MCAN + Hami-m³d | 92.01±0.80* | 92.01±0.80* | 90.44±0.70* | 93.37±0.87* | 91.88±0.85* | 93.59±0.74* | 90.84±0.78* | 92.17±0.76* | +0.98 |
| CAFE [8] | 90.99±0.78 | 90.98±0.78 | 90.31±0.72 | 91.19±1.09 | 90.73±0.97 | 91.70±1.26 | 90.81±1.03 | 91.24±0.60 | - |
| CAFE + Hami-m³d | 91.95±1.06* | 91.84±1.01* | 91.25±0.55* | 92.38±1.04* | 91.66±0.91* | 92.99±0.83* | 91.93±0.91* | 92.11±0.75* | +1.02 |
| BMR [52] | 90.17±0.92 | 90.15±0.93 | 90.09±1.20 | 89.60±0.85 | 89.81±1.00 | 90.36±0.93 | 90.71±0.78 | 90.50±0.81 | - |
| BMR + Hami-m³d | 91.74±0.40* | 91.68±0.40* | 91.01±0.92* | 93.17±0.82* | 91.56±0.43* | 93.40±0.84* | 91.29±0.67* | 91.81±0.38* | +1.79 |
| **Dataset: *Twitter*** | | | | | | | | | |
| Basic model | 65.08±1.18 | 63.91±1.09 | 57.29±1.26 | 66.67±1.01 | 61.48±1.56 | 72.04±0.96 | 62.41±0.92 | 65.35±1.01 | - |
| Basic model + Hami-m³d | 66.27±0.66* | 65.67±1.27* | 59.70±1.16* | 69.70±0.71* | 62.46±1.08* | 73.19±0.93* | 64.12±1.12* | 67.86±0.82* | +1.84 |
| SAFE [60] | 66.43±0.33 | 66.33±0.32 | 58.28±0.50 | 73.63±1.38 | 64.47±0.53 | 74.94±0.84 | 61.78±1.26 | 68.34±0.69 | - |
| SAFE + Hami-m³d | 67.15±0.96* | 67.00±0.89* | 59.32±0.90* | 74.05±0.99 | 65.65±0.70* | 76.49±0.60* | 63.58±1.09* | 68.77±0.94 | +0.98 |
| MCAN [51] | 65.82±0.64 | 65.24±1.34 | 58.30±1.07 | 63.66±1.03 | 61.16±1.23 | 71.70±1.03 | 67.42±1.39 | 69.33±1.22 | - |
| MCAN + Hami-m³d | 67.14±1.11* | 66.58±1.21* | 60.63±0.99* | 64.94±1.04* | 62.55±1.28* | 72.86±0.82* | 68.77±1.12* | 70.61±1.10* | +1.43 |
| CAFE [8] | 65.62±0.58 | 65.04±0.48 | 58.39±0.90 | 66.24±1.48 | 62.05±0.21 | 72.37±1.28 | 65.16±1.06 | 68.57±1.05 | - |
| CAFE + Hami-m³d | 65.89±1.30 | 65.37±0.87 | 59.91±0.55* | 67.28±1.17* | 63.60±0.64* | 73.42±1.18* | 68.76±1.12* | 70.49±1.06* | +1.41 |
| BMR [52] | 67.12±0.74 | 66.64±1.28 | 59.09±0.61 | 72.62±1.28 | 64.43±1.28 | 75.10±1.13 | 62.56±0.91 | 68.65±1.17 | - |
| BMR + Hami-m³d | 67.84±0.83* | 67.68±0.82* | 60.01±0.88* | 73.31±1.28* | 65.65±0.92* | 76.27±1.03* | 64.32±0.98* | 69.71±0.91* | +1.08 |

- **BMR** [52] creates an elaborate network with improved mixture-of-experts to extract and fuse multimodal features.

The results of all baselines are re-produced by us to use the BERT model and ResNet34 as the feature extractors.

**Implementation Details.** To preprocess the raw data, we resize and randomly crop raw images to $224 \times 224$, and truncate text content to 128 word tokens. Then, we employ pre-trained ResNet34 and BERT[3] to capture visual and text features, and the first 9 Transformer layers of the BERT model are frozen. For the manipulation teacher, we use a shallow ResNet18 model as its backbone, and we pre-train it with an existing benchmark dataset *CASIAv2*[4] [14, 31] on image manipulation detection, which consists of 12,614 images, including 7,491 authentic and 5,123 tampered images. During training, we fine-tune the BERT model using an Adam optimizer with a learning rate of $3 \times 10^{-5}$ and optimize the other modules using Adam with a learning rate of $10^{-3}$, and the batch size is consistently

fixed to 32. We empirically set the hyperparameters $\alpha$, $\beta$, $\delta$, and $K$ to 0.1, 0.1, 0.1, and 10, respectively. Meanwhile, to prevent overfitting, we stop the training early when no better Macro F1 score appears for 10 epochs.

## 4.2 Main Results

We compare the performance of our model Hami-m³d against 5 baseline models across 3 benchmark datasets, and evaluate them using 8 typical metrics. The experimental results are reported in Table 2. Generally, Table 2 reports the scores of Avg. Δ, which represents the average improvements of Hami-m³d over the baseline models across all evaluation metrics. We observe significant improvements of Hami-m³d over all these baselines. For instance, on the *Weibo* dataset, Hami-m³d improves the BMR model by approximately 1.79, and on the *Twitter* dataset, it improves the basic model by 1.84. Observing the performance of Hami-m³d in detail on different metrics, our model consistently outperforms the baseline models on all evaluation metrics. For example, on the *Gossipcop* dataset, it outperforms the BMR model by approximately 2.57 in

---

[3]Downloaded from https://huggingface.co/bert-base-uncased.

[4]Downloaded from https://github.com/SunnyHaze/CASIA2.0-Corrected-Groundtruth.

**Table 3: Ablative study on objective functions. w/o represents the short of "without". The bold and underlined scores indicate the highest and lowest results in the ablative versions, respectively.**

| Method | Acc. | F1 | AUC | F1$_{\text{real}}$ | F1$_{\text{fake}}$ |
|---|---|---|---|---|---|
| **Dataset: *GossipCop*** | | | | | |
| BMR + Hami-m³d | 87.95 | 79.99 | 87.46 | 93.14 | 66.80 |
| w/o $\mathcal{L}_{PRE}$ | 87.24 | 79.18 | 87.21 | 92.14 | 66.23 |
| w/o $\mathcal{L}_{PU}$ | **87.63** | **79.66** | **87.38** | **93.05** | **66.52** |
| w/o $\mathcal{L}_{KD}, \mathcal{L}_{IR}$ | 86.93 | 78.65 | 86.24 | 91.94 | 65.36 |
| **Dataset: *Weibo*** | | | | | |
| BMR + Hami-m³d | 91.74 | 91.68 | 97.06 | 91.56 | 91.81 |
| w/o $\mathcal{L}_{PRE}$ | 90.17 | 90.12 | 96.93 | 89.91 | 90.83 |
| w/o $\mathcal{L}_{PU}$ | **91.40** | **91.39** | **96.96** | **91.19** | **91.60** |
| w/o $\mathcal{L}_{KD}, \mathcal{L}_{IR}$ | 90.10 | 90.10 | 95.97 | 89.88 | 90.31 |

**Table 4: Ablative study on manipulation and intention features. w/o represents the short of "without".**

| Method | Acc. | F1 | AUC | F1$_{\text{real}}$ | F1$_{\text{fake}}$ |
|---|---|---|---|---|---|
| **Dataset: *GossipCop*** | | | | | |
| BMR + Hami-m³d | 87.95 | 79.99 | 87.46 | 93.14 | 66.80 |
| w/o $\mathbf{e}^M$ | 87.53 | 79.13 | 86.47 | 92.37 | 65.89 |
| w/o $\mathbf{e}^E$ | **87.69** | **79.56** | **86.72** | **92.39** | **66.25** |
| w/o $\mathbf{e}^M, \mathbf{e}^E$ | 87.26 | 79.03 | 86.27 | 92.14 | 65.51 |
| **Dataset: *Weibo*** | | | | | |
| BMR + Hami-m³d | 91.74 | 91.68 | 97.06 | 91.56 | 91.81 |
| w/o $\mathbf{e}^M$ | 90.92 | 90.91 | 96.52 | 90.57 | 91.08 |
| w/o $\mathbf{e}^E$ | **91.13** | **91.11** | **96.78** | **90.78** | **91.45** |
| w/o $\mathbf{e}^M, \mathbf{e}^E$ | 90.17 | 90.15 | 96.45 | 89.81 | 90.50 |

the fake class recall score. These results demonstrate the effectiveness of our approach and highlight the role of manipulation and intention features in detecting misinformation. Additionally, we observe that the order of improvements induced by Hami-m³d across the three datasets is roughly ranked as *Twitter* > *Weibo* > *GossipCop*. This phenomenon suggests that, firstly, for smaller datasets, Hami-m³d compensates for the lack of semantic information by exploiting manipulation and intention features, leading to larger improvements. Secondly, we observe that there are more manipulated images in the *Twitter* dataset, allowing our manipulation features to have a greater impact, which indirectly demonstrates the accuracy of our extracted manipulation features.

## 4.3 Ablative Study

To evaluate the effectiveness of all objective functions and features in Hami-m³d, we conduct an ablative experiment on an English dataset *GossipCop*, and a Chinese dataset *Weibo*, their experimental results are shown in Tables 3 and 4. The descriptions of these ablative versions are as follows:

- **w/o** $\mathcal{L}_{PRE}$ represents the removal of the pre-training process using the external IMD dataset $\mathcal{D}_\mu$, training the teacher model only with the PU objective $\mathcal{L}_{PU}$;
- **w/o** $\mathcal{L}_{PU}$ indicates not using the PU loss to adapt the teacher model to the MMD dataset $\mathcal{D}$;

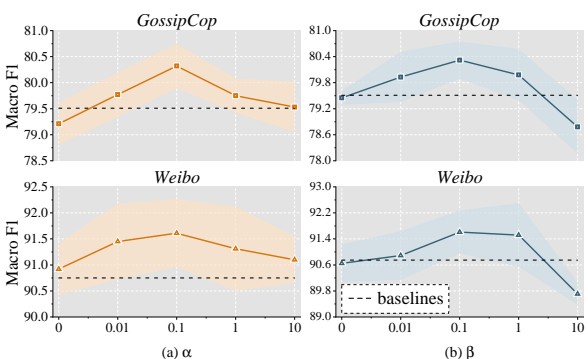

**Figure 3: Sensitivity analysis of the parameters $\alpha$ and $\beta$.**

- **w/o** $\mathcal{L}_{KD}, \mathcal{L}_{IR}$ represents not using any objective function to supervise the manipulation and intention features, and the optimization of these two features solely relies on the cross-entropy loss $\mathcal{L}_{CE}$ of veracity prediction. Due to the strong dependency of $\mathcal{L}_{IR}$ on $\mathcal{L}_{KD}$, we exclude both of them in this analysis;
- **w/o** $\mathbf{e}^M$, **w/o** $\mathbf{e}^E$, and **w/o** $\mathbf{e}^M, \mathbf{e}^E$ represents the removal of $\mathbf{e}^M$ and $\mathbf{e}^E$ and their related training losses, as well as the removal of both features, equivalent to the baseline model without using the method proposed in this paper.

In general, removing each module leads to a decrease in the prediction results of Hami-m³d, confirming their effectiveness. Specifically, comparing the ablation results of the three objectives, their performance is roughly ranked as w/o $\mathcal{L}_{PU}$ > w/o $\mathcal{L}_{PRE}$ > w/o $\mathcal{L}_{KD}, \mathcal{L}_{IR}$, and the removal of $\mathcal{L}_{PRE}$ and $\mathcal{L}_{KD}, \mathcal{L}_{IR}$ has the greatest impact on the results of our model, with several results even falling below the baseline model. $\mathcal{L}_{PRE}$ effectively guides the manipulation teacher to accurately determine whether an image has been manipulated. Without $\mathcal{L}_{PRE}$, the teacher's prediction performance deteriorates, indirectly leading to more confused features obtained by the manipulation encoder, thereby significantly affecting the veracity prediction results. On the other hand, $\mathcal{L}_{KD}$ and $\mathcal{L}_{IR}$ impose no constraints on the manipulation and intention features, not only increasing the computational burden of the baseline model but also introducing meaningless and non-discriminative features, directly affecting the discriminative ability of the final veracity feature $\mathbf{e}$. Then, comparing the ablation results of the three features, their performance is ranked as w/o $\mathbf{e}^E$ > w/o $\mathbf{e}^M$ > w/o $\mathbf{e}^M, \mathbf{e}^E$, indicating that both features contribute to enhancing the discriminative ability of the final multimodal feature, and $\mathbf{e}^M$ has a greater impact.

## 4.4 Sensitivity Analysis

In Hami-m³d, $\alpha$ and $\beta$ are crucial hyper-parameters, which represent the weights of $\mathcal{L}_{KD}$ and $\mathcal{L}_{IR}$ to balance training among multiple losses. Therefore, in this section, we conduct sensitivity experiments on these two hyper-parameters to analyze whether our model is sensitive to these parameters and to provide evidence for the selection of hyper-parameters in Hami-m³d. The specific experimental results are shown in Fig. 3. We conduct experiments across

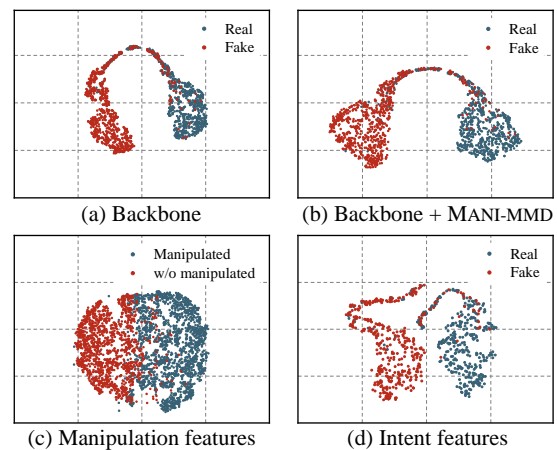

(a) Backbone  (b) Backbone + MANI-MMD

(c) Manipulation features  (d) Intent features

**Figure 4: Visualization analysis of features z, $e^M$ and $e^E$ with the T-SNE method.**

both English and Chinese datasets, *GossipCop* and *Weibo*, respectively, and report the Macro F1 metric in Fig. 3. $\alpha$ and $\beta$ are selected from the set {0, 0.01, 0.1, 1, 10}, where $\alpha$ or $\beta = 0$ indicates that the corresponding objective function does not need to be trained. The experimental results show that the model is quite sensitive to both hyper-parameters and consistently has the best performance when $\alpha = 0.1$ and $\beta = 0.1$. As they increase or decrease, the model's performance shows a decreasing trend. Therefore, in implementing all the experiments in this paper, we always choose $\alpha = 0.1$ and $\beta = 0.1$. When $\alpha$ and $\beta$ are small, the manipulation and intention features are not sufficiently trained, leading to insufficient discriminative features that degrade the model's prediction results, even below the baseline model. Conversely, when they are large, the model's optimization tends to favor their corresponding losses, reducing the weight of the veracity prediction objective $\mathcal{L}_{CE}$ and degrading the veracity prediction results.

## 4.5 Visualization Analysis

To analyze the discriminative nature of the extracted manipulation and intention features, we visualize these features, as shown in Fig. 4. Specifically, we choose the *Weibo* dataset for visualization analysis, using the T-SNE method [41] to reduce the dimensionality of the multimodal feature z, manipulation feature $e^M$, and intention feature $e^E$ to 2D, and displaying the corresponding 2D points in Fig. 4. Fig. 4(a) illustrates the visualization of the multimodal features of the basic model, while Fig. 4(b) shows the visualization of the multimodal features with the addition of our proposed HAMI-$M^3D$. By comparing the two results, we can observe that our method can separate the two clusters of *real* and *fake* classes from each other, thereby improving the discriminative nature of the multi-modal features to a certain extent. Fig. 4(c) displays the visualization results of the manipulation feature $e^M$, where we use the results provided by the teacher model to distinguish between *manipulated* and *unmanipulated* images. In the result, we can observe that the manipulation feature has strong discriminative power in determining whether an image has been manipulated, demonstrating the

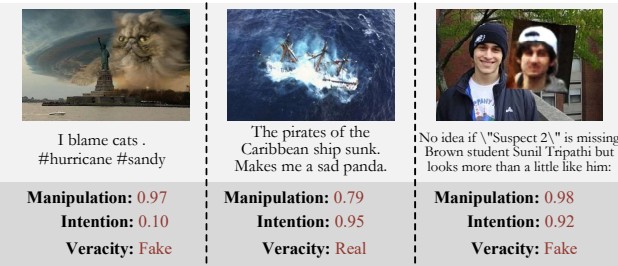

**Figure 5: We illustrate three representative examples for the case study.**

effectiveness of knowledge distillation in inheriting the discriminative ability of the manipulation teacher to the manipulation encoder. Fig. 4(d) visualizes only the intention features $e^E$ of samples classified as "*manipulated*" in the manipulation classification task. We find that the intention features of *real* and *fake* samples are clearly separated. However, some *fake* samples are mixed in the *real* cluster, reflecting the fact that *fake* samples may also have a *harmless* intent.

## 4.6 Case Study

We provide three representative examples for illustrating the performance of our classifiers on three tasks in Fig. 5. The first example involves an image that has been manipulated in a harmful intention. Our model confidently predicts both the manipulation and the intention, accurately identifying its veracity label as fake; The second example presents an image with manipulated colors. Although our model correctly identifies the manipulation and provides accurate veracity predictions, it assigns a low-confidence probability to this type of manipulation, suggesting that there is room for improvement in detecting certain manipulation techniques; The third example illustrates an image that has been harmlessly manipulated, merely slicing it without any harmful intention. Our model also provides accurate predictions for all three tasks. In summary, our model's performance across the three tasks is commendable, but it is susceptible to subtle manipulations that require further improvement.

## 5 CONCLUSION

In this paper, we aim to identify multimodal misinformation by recognizing manipulation traces of images in articles, as well as understanding the underlying intention behind such manipulation. To this end, we introduce a novel MMD model named HAMI-$M^3D$, which extracts manipulation and intention features and incorporates them into the overall multimodal features. To make manipulation and intention features discriminative towards whether the image has been harmfully manipulated, we propose two classifiers and predict their respective labels. To address unknown manipulation and intention labels, we propose two weakly supervised signals by learning a manipulation teacher with additional IMD datasets and using two PU learning objectives to adapt and supervise the classifier. Our experimental results can demonstrate that HAMI-$M^3D$ can significantly improve the performance of its baseline models.

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
