# OpenReview forum: "Harmfully Manipulated Images Matter in Multimodal Misinformation Detection"
_acmmm.org/ACMMM/2024/Conference — MM2024 Poster_

### Official Review · Reviewer_5HFG · 2024-05-21

**Rating:** 3
**Confidence:** 3

**Summary:**

The authors propose to detect misinformation by learning manipulation features that indicate whether the image has been manipulated, as well as intention features regarding the harmful and harmless intentions of the manipulation and introduce the model Harmfully Manipulated Images Matter ($HAMI-M^3D$). To address the challenge of unknown manipulation and intention labels,  $HAMI-M^3D$ employs two weakly supervised signals by learning a manipulation teacher with additional Image Manipulation Detection datasets and using two Positive-Unlabeled (PU) learning objectives to adapt and supervise the classifier.

**Strengths:**

1. The authors creatively tackle the issue of unavailable manipulation and intention labels. This innovative attempt is noteworthy and could inspire further research in the field.
2. The paper is well-organized and adheres to academic writing standards. The structure is complete, making the research easy to follow and understand.
3. The authors conducted extensive experiments across three misinformation detection datasets to validate the performance of the proposed model.

**Limitations:**

1. **Label Definition Ambiguity:** There seems to be a contradiction in the definition of labels (indicating 1 or 0). In lines 510-512, $y_i^E=1$ indicates harmless, while in lines 537-539, $y_i^E=1$ indicates harmful. Clarifying this ambiguity is essential for better understanding.
2. **About Error Propagation:** There is a potential issue where errors in the manipulation detection part might propagate to the intention detection part. How do the authors address this problem?
3. **Experimental Results:** The improvements in experimental results are not very significant, including those from ablation studies. This could be due to the relatively small ratio of image manipulation data in the datasets used. The authors are encouraged to test their model on datasets with more manipulated images to validate its efficacy further.
4. **PU Learning Outcomes:** Providing the final proportions of manipulation and intention labels in real and fake data (determined by the PU learning approach) would be useful for understanding the data condition.
5. **Typos:** There are several typos in the paper, such as $L_{VC}$ being referred to as $L_{VP}$ in Figure 2. Correcting these would enhance the manuscript's clarity.
6. **Code and Data Availability:** The authors have not provided their code and data. Making these resources available would significantly benefit the research community by enabling replication and further study.

**Suitability:**

2

---

### Official Review · Reviewer_k9Tj · 2024-05-23

**Rating:** 4
**Confidence:** 3

**Summary:**

This paper focuses on the task of multimodal misinformation detection and introduces a novel approach by incorporating manipulation and its intent as key components of this task. This paper proposes the HAMI-M^3D model, which includes manipulation and intent classification as auxiliary tasks. These are treated as positive and unlabeled (PU) learning problems. Extensive experiments are conducted to demonstrate the effectiveness of the proposed method.

**Strengths:**

1. The paper presents an intriguing research perspective in the context of MMD-related studies. The relationship between veracity labels, manipulation labels, and intention labels is intuitively complex. This article innovatively models these as a problem of positive and unlabeled (PU) learning, offering valuable insights that could inspire further research in the field.
2. The experiments are extensive, testing the proposed model against five baselines across three datasets. The authors also conduct a detailed analysis of parameter sensitivity and provide interesting visualization, enhancing the understanding of the model’s dynamics and its performance.

**Limitations:**

**Motivation:**
- Despite the significant link between image manipulation and the authenticity of news, image tampering in most real-world scenarios and MMD datasets is relatively rare. I have reservations about the claim in the paper that "approximately 66.4% of fake articles involve manipulated images" observed in the MediaEval15 dataset. This figure may not be representative and could mislead readers. I recommend that the authors conduct a prior analysis of other datasets or revise this claim to prevent potential misguidance.

**Experiments:**
- The proposed method only improves the average performance of its baselines by approximately 1.21% across all metrics, which I’m afraid may not be considered effective given the variability inherent in deep learning.
- The paper mentions that the GossipCop dataset comprises 12,840 items, which differs from the original count of 18,417. Clarification is needed on how this number was derived.
- The ratio of fake to real news in the GossipCop dataset is close to 1:10. How was this imbalance addressed? Was this ratio maintained across train, validation, and test splits?
 - What settings were used to repeat experiments to obtain the reported means and variances?
 - In the case study, why were the second and third examples both interpreted as harmless tampering, yet one was classified as real and the other as fake?

**Other Necessary Revisions:**
- Ensure correct citations, including referencing the Twitter dataset upon first mention; also, the part of the GossipCop dataset mentioned is actually from the FakeNewsNet dataset and should be referred to as such.
- It is advisable to label intent as 0/1 to clarify which class they correspond to, avoiding confusion.
- The sub-title in Figure 4(b), "MANI-MMD," should be corrected to "HAMI-M^3D."

**Suitability:**

3

---

### Official Review · Reviewer_WJBA · 2024-05-25

**Rating:** 5
**Confidence:** 3

**Summary:**

This is an article about multimodal misinformation detection. The authors start from manipulated traces of the images based on intuition and statistical analysis to uncover image manipulation and detect misinformation. This approach aligns with intuitive thinking and is supported by statistics, making it a reasonable and feasible method for multimodal misinformation detection.

**Strengths:**

1.The intuitive thinking behind the method is not particularly novel, but it is very reasonable. It aligns with intuitive cognition and is supported by statistical information. Figure 1 illustrates the motivation for this method very clearly and intuitively.

2.The Hami-m3d framework is clear and integrates three classification sub-tasks. Notably, it is not an independent network and can be combined with current multimodal misinformation detection networks to further enhance performance. This is clearly demonstrated in the results shown in Table 2.

3.The experimental results in the paper are comprehensive and validate the proposed method. Table 2.

**Limitations:**

1.Although manipulation detection based on statistical intuition is effective and reasonable for identifying virtual information, it does not seem particularly unique. Some tasks might also use similar techniques for tampering detection and face modification/generation detection to enhance their results.

2."The image of the real information has been manipulated, its intention must be harmless", is such manipulation for purposes like beautification? Based on this idea, does the design of this method rely entirely on the introduced IMD dataset for training the teacher model?
The most critical aspects of manipulation detection and intent recognition heavily rely on the teacher model trained on the IMD dataset. It appears there are no additional special designs beyond this.

3. Some manipulations seem to easily mislead the model into categorizing real images entirely as misinformation. It seems more reasonable to add some noise layers to the teacher model.

**Suitability:**

3

---

### Official Review · Reviewer_a7hR · 2024-05-27

**Rating:** 4
**Confidence:** 3

**Summary:**

Existing multimodal misinformation detection(MMD) methods often concentrate on the semantic associations and inconsistencies between multimodal contents, but they tend to neglect the potential clues found in traces of image manipulation. The fact that an image has been manipulated and the underlying intentions of such manipulation, whether harmful or harmless, are of paramount importance for detecting misinformation. The paper introduces a novel approach known as Harmfully Manipulated Images Matter in MMD (Hami-m3d), designed to detect misinformation on social media platforms that contains manipulated images within multimodal content.
The method begins by extracting semantic features from textual and visual content through feature encoders, followed by the utilization of a manipulation encoder and an intention encoder to separately derive features that indicate the presence of image manipulation and the harmfulness of the intentions behind it. Given that direct labels for manipulation and intentions are typically unavailable, Hami-m3d leverages weakly supervised signals, employing knowledge distillation to train a manipulation teacher model, and integrates a Positive and Unlabeled (PU) learning framework to adapt and supervise the classifiers. Operating under a multi-task learning framework, Hami-m3d concurrently refines the primary task of veracity classification along with auxiliary tasks of manipulation and intention classification, leading to a marked improvement in the performance of baseline models across various benchmark datasets.
Comprehensive experiments conducted across three benchmark datasets have demonstrated that Hami-m3d can consistently elevate the efficacy of existing MMD baseline models.

**Strengths:**

This article proposes a method called Hami-m3d, which introduces a new dimension to multimodal misinformation detection by combining manipulation features and intention features to detect image-manipulated misinformation.
Utilizing weakly supervised learning and knowledge distillation techniques, it addresses the issue of missing labels, offering a comprehensive solution to enhance the accuracy of misinformation detection.

**Limitations:**

The introduction of the Hami-m3d method is not clear. The model distillation part requires a more detailed explanation.
The experiments only compared with a relatively new fake news detection method BMR and some older methods, and did not compare with other methods from recent years, which makes the comparison scheme less convincing.
There are several blank lines at 164-166, 454-457, and the layout here can be improved. Figure 2, the core framework diagram, can be more balanced and needs to be drawn more beautifully.

This paper incorporates manipulation intent into the MMD task and enhances model performance using weakly supervised signals and knowledge distillation, which has a certain degree of innovation. However, there are some shortcomings in the overall layout and drawing that need improvement. In addition, the experiments should include not only comparisons with classic models but also with the latest models.

**Suitability:**

2

---

### Meta-Review · Area_Chair_6cgH · 2024-07-02

**Recommendation:** Accept (Poster)
**Confidence:** 5

**Metareview:**

This paper considers the multi-modal misinformation detection task. The authors give a special focus on harmfully manipulated images and propose to detect fake news by learning manipulation features that indicate whether the image has been manipulated. All reviewers provide positive final ratings, considering that this paper provides a new perspective and a reasonable technical solution. I recommend the acceptance of this paper, but the authors are strongly required to revise this paper carefully according to the reviews to further improve the paper's quality.